# Strategic rule breaking: Time wasting to win soccer games

**Henrich R. Greve** [1]* , **Nils Rudi** [2] , **Anup Walvekar** [3]

**1** INSEAD, Singapore, Singapore, **2** Yale School of Management, New Haven, CT, United States of America, **3** National University of Singapore Business School, Singapore, Singapore

☉ These authors contributed equally to this work.
* henrich.greve@insead.edu

## Abstract

Rules regulate behavior, but in competitive contexts they also create incentives for rule-breaking because enforcement is imperfect. Sports is a prime example of this, and one that lends itself well to investigation because strategic rule-breaking is often measurable. Professional soccer is a highly competitive team sport with economic rewards for winning given to teams and players. It has a set of rules to ensure fair play, but the enforcement is incomplete, and hence can lead to strategic behavior. Using newly available data, we examine strategic time-wasting, a behavior that help teams win games, or tie games against superior opponents, but is contrary to the objective of game play as entertainment for the spectators. We demonstrate that strategic time-wasting is widespread and is done through delayed restart of the game after goalie capture of the ball, goal kick, throw-in, free kick, corner kick, and substitution. The strategic time-wasting has substantial magnitude, and models of the value per minute predict time-wasting well. Because this time-wasting is a result of incentives created by not stopping the game clock, we predict that a change to rules with stopped game clock when the play is stopped would make game play more time efficient.

## Introduction

Most arenas of social life have rules, ranging from norms for good behavior to regulations and laws with enforcement and penalties. The same areas of social life often have benefits obtained through competition, with potential to gain advantages through breaking the rules. Indeed, one purpose of rule systems is to distinguish acceptable and unacceptable competitive actions in order to increase social welfare [1]. A prototypical case is professional sports, which have clearly defined sets of rules and a lengthy track record of rule-breaking, in some cases providing great rewards to the perpetrator. Professional cyclists have engaged in significant and systematic doping [2], with the profits earned by Lance Armstrong alone estimated to USD 200 million. Manipulation of equipment is also done, as in the "deflategate" leading to National Football League's New England Patriots receiving a fine and losing draft picks after they were found to have deflated the football to advantage their quarterback. Even the game field can be manipulated, as when ice hockey players deliberately move the goal when under attack.

(jens.melvang@prozonesports.com). The authors
did not have special access privileges.

**Funding:** The author(s) received no specific
funding for this work.

**Competing interests:** The authors have declared
that no competing interests exist.

The breaking of rules in competitive arenas is closely related to theory of rational crime and optimal law enforcement [3, 4]. Holding the benefit of rule breaking constant, it is more likely to occur when the enforcement, defined as the product of the discovery likelihood and punishment severity, is lower. Holding the enforcement constant, it is more likely to occur when the benefit is greater. Research on rule breaking in and by organizations has cast doubt on whether rule breaking is fully rational, however, because other factors have been found to matter, such as psychological strain, cultural beliefs, and interpersonal influence [5]. Because it is usually difficult to measure rule-breaking opportunities and benefits using field data, accurate answers to the question of whether rule breaking is strategic and, if so, how the strategy is formed, are rarely found.

We turn to soccer as an excellent empirical context for investigating this question. Soccer is the world's largest sports in revenue, viewers, and athletes. The UEFA Champions League has 2.28B USD in prize money, the highest of any sports competition, and the top 5 highest paying sports competitions in terms of prize money, includes 3 other soccer competitions. Underlying the prize money are expensive broadcasting rights, such as the 5.3B GBP paid for the domestic broadcast of English Premier League. The broadcasting rights are expensive because of the high audience interest, as soccer is the most-watched sport in many nations, and the two last World Cup competitions each had more than 3B TV viewers. The number of people who regularly play soccer is around 265M worldwide. www.totalsportek.com/most-popular-sports/ summarizes these statistics.

Football has a detailed rule system that has been largely unchanged since it was formalized in the 19th century. Unlike many other sports, the game clock is not stopped when the play halts because the ball goes out of bounds or there is a foul or substitution. If there had been no strategic time-wasting, the loss of playing time would just be a random result of the events in the game. Indeed, to prevent loss of time from being used strategically, time-wasting is a foul that should be penalized by the referee. Strategic time-wasting is still done by delaying restart of play after game events that involve stoppage to retrieve and position the ball and players (goal kick, throw-in, free kick, and corner kick) or temporary holding of the ball to allow teammates to reposition themselves (goalie kick or throw of captured ball). The purpose of time-wasting is to secure a win when the team is ahead in the score, or even to secure a tie when the score is even and the opposing team is stronger.

Time-wasting is recognized as a potential problem by fans of the game and the soccer governing bodies. There are discussions in the international football association FIFA and the governing body IFAB of whether the current rules give too much room for strategic time-wasting, and hence need to be changed. Any changes in the rules, such as to stop the clock when the ball is out of play, would be a major break with the traditional way of playing the game, and would not be easy to make without clear evidence that time-wasting is significant in duration and could potentially affect game results. These are questions that can be answered by using management science techniques for analyzing the duration of game stoppages, as we do here. Indeed, the discussions at FIFA have been informed by findings from our research project.

Fig 1 can be used to assess the extent of strategic time-wasting. For each stoppage event, it compares stoppages in which the restarting team is ahead, tied, and behind in goals scored. The heat maps show the number of stoppages normalized to 60 minutes of in-play time (horizontal axis) and the duration of each stoppage in seconds (vertical axis). For all events except substitutions, the number of stoppages is determined by the course of play, with the team restarting the play having some ability to create stoppages, but often it is the team not restarting that creates stoppages. Thus, while movements North or East in the heat map means more time wasted, a movement North is more indicative of the restarting team's influence. The histograms on each axis depict the distribution along the corresponding dimension, so the

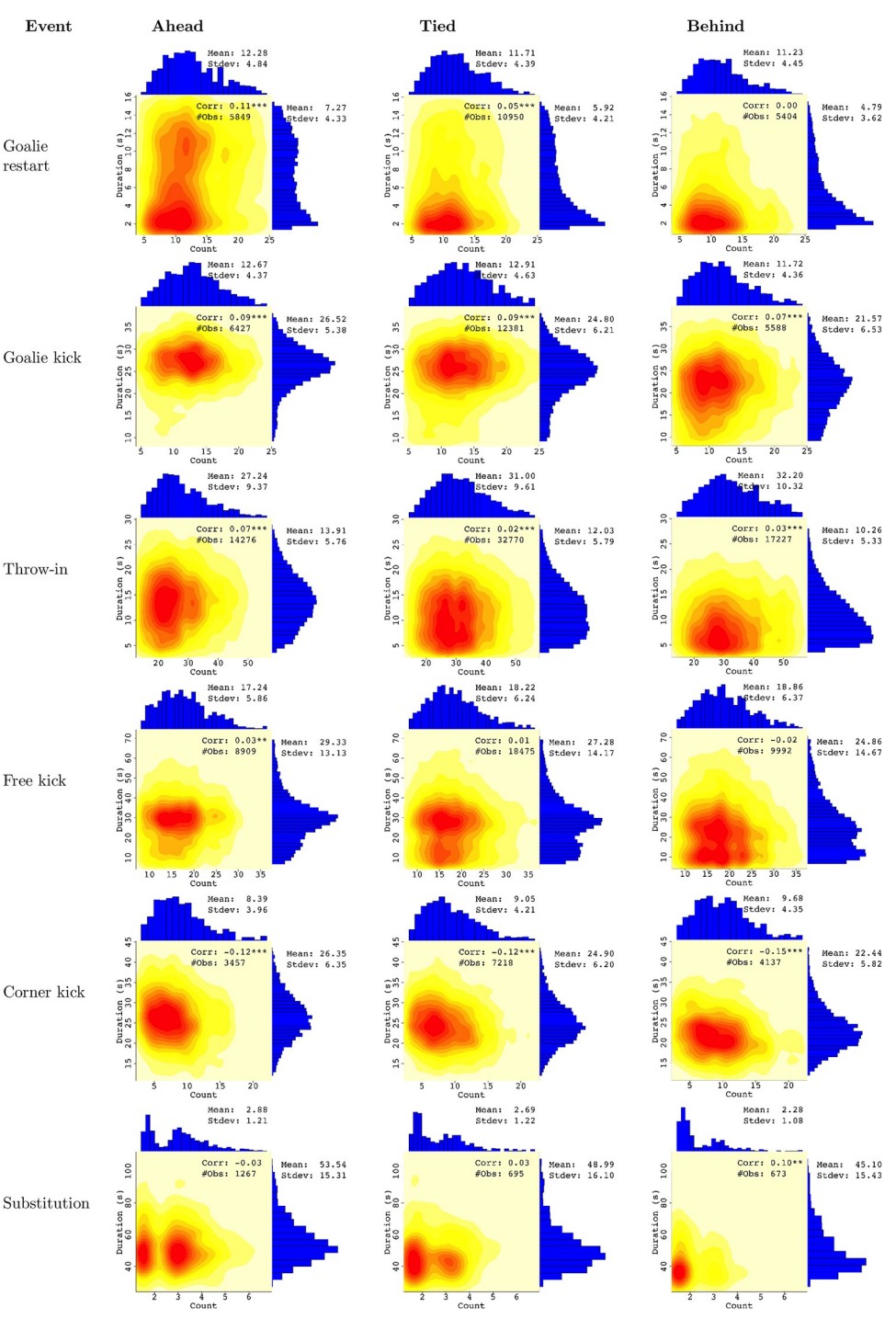

**Fig 1. Frequency and duration of each stoppage.**

vertical (rightmost) histogram shows duration and is of particular interest for assessing time-wasting.

Fig 1 shows that goalie restart of play after capturing the ball in play has many more lengthy events when the team is ahead. Goal kicks are not visibly different when the restarting team is ahead or tied (the mean is slightly larger when ahead) but are faster when it is behind.

Consistent with strategic time-wasting, throw-ins are faster when the team is tied, and even faster when it is behind. Free kicks are similar: the duration is reduced as the restarting team goes from ahead to tied, and from tied to behind. Corners have longer duration when the restarting team has a less advantageous score. Throw-ins, free kicks, and corners are less frequent when the restarting team is ahead, which could be a natural result of the flow of play or a result of the team behind seeking to keep the ball in play. Finally, for substitutions the heat map shows movement towards Northeast when the restarting team is ahead in the score during the last 15 minutes of the game. We omit earlier substitutions, which are usually made to replace a player who is injured or has played poorly. This is because the team that calls for more substitutions also controls the duration. The scarcity and high speed of substitutions by the team that is behind are particularly striking.

Although this figure demonstrates strategic time-wasting, splitting the data into scores when the restarting team is ahead, tied, or behind is a crude approach that ignores multiple features of the game. First, the teams have unequal strength even in a top league, and a tied score has different meaning for two teams that have roughly the same strength and two teams with very different strength. A tied score is advantageous for the weaker of two teams, a fact that Fig 1 ignores. Second, the game of soccer has a nominal time of 90 minutes, and the effect of wasting time on the final result is very different depending on the time at which a stoppage occurs. Again, the weaker of two tied teams can accomplish little by wasting time in minute 10 of the game, but time-wasting in minute 80 could be very valuable. The division into ahead, tied, and behind, or indeed any score difference, is a simple heuristic that may underestimate the strategic nature of time-wasting.

To analyze the time-wasting strategy, we use a value-based model of time-wasting benefit as a reference point. We apply a model for in-play prediction of match outcomes [6] which is briefly described as follows. For each match, let $i$ index each team and let $j$ index the match outcome (team 1 wins, tie, team 2 wins), where the probability of outcome $j$ is denoted $p_j$. The goals of the two teams are modeled as two independent Poisson processes with arrival rates $\lambda_i$, where the number of goals scored by team $i$ up to time (in minutes) $t$ is denoted $N_i(t)$. With this, the probability at time $t$ of an outcome given current score difference $s(t) = N_1(t) - N_2(t)$ can be calculated. For an arbitrary pre-match predictive distribution of team strengths, there exists a unique pair of Poisson arrival rates $\lambda_1$, $\lambda_2$ corresponding to the above described model that in turn can be used to estimate the probability of the outcomes (team 1 wins, tie, team 2 wins) given any elapsed time of the game and score [6]. Below we describe how this model yields variables that predict strategic time-wasting.

In addition to the value-based model, we also use a semi value-based model assuming that the pre-game team strength is taken into account and the probability is adjusted for game state and time remaining as described in the psychological theory of anchoring and adjustment [7]. We fit these models to the data and produce parameter estimates and model fit.

The models show that soccer players strategically waste time, which means that the 90 minutes of nominal play time is reduced not only by natural stoppages such as fouls or the ball out of bounds, but also by teams strategically taking advantage of such stoppages. This suggests the potential for changing the rules of the game to one with a shorter effective play time where the game clock is stopped whenever play is halted, and such a rule change is currently discussed by IFAB/FIFA. Under current rules, effective play time is significantly less than 60 minutes per game, so this change of rules would lead to more effective play time. Because the incentive to prolong stoppages would disappear, the stoppage time would be reduced. The result would be an increase in the proportion of game time with the ball in play, to the advantage of game spectators. But even more importantly, the variability of play time would be eliminated, it would

become a more fair game, and unethical behavior of players would be reduced, the latter of which is a concern because they are idols forming norms for children.

## Materials and methods

### Data

We study the top division of one of the top professional soccer leagues in Europe. It has 20 teams and plays according to a league (round-robin) system to determine a champion and three teams that will be relegated to the division below. The top four teams will advance to the Europe-wide Champions League, so the final position on the table determines whether the team will end the season in either of the three tiers of the league: Champions League, middle, or relegation.

We analyze data from Amisco-Prozone, which codes soccer matches from many professional leagues for analytical use by teams. The data contain player position on the field (x, y coordinates) every 0.1 seconds and events, which are coded from video recordings and classified into 58 types. Events can be as basic as a touch (player touches the ball, usually to drive it forward), or as specific as a goalie punch (with the fists), fumble (open hands, dropped ball), catch (open hands and holding) and save (shot block that prevented a goal). On average a game has 2,557 events. We use data from 2,231 matches from the 2009–2015 seasons.

The events of interest here stop the play and allow one team to delay the restart of play. We focus our analysis on common events, and hence we analyze the same events as shown in Fig 1. They are (1) goalie kick or throw after capturing the ball, (2) goalie kick after the opposing team has made the ball cross the goal-line, (3) throw-in after the opposing team has made the ball cross the touchline, (4) corner kick, (5) free kick and (6) substitution. Events 1 and 2 involve the goalie, whose unique control over the ball may make him a designated time-waster, and event 6 involves the manager. The other events involve regular team members.

Time-wasting should be seen as a game among members of the team controlling the ball, members of the opposing team, and the referee team. Referees are required to prevent time-wasting and to penalize it with yellow cards. When players of the team that will not restart the game are near the ball as the referee stops the play, they can try to speed up play by quickly returning the ball to the throw-in or free kick spot, or to slow it down by being slow to return the ball. The latter is a behavior that referees are especially alert to and may penalize. The incentives in favor of time-wasting could still influence a team to waste time despite referee and opposing team's attempts to limit time wasting.

### Models

In each analysis of time-wasting events, the dependent variable is the duration from the game stoppage to restart, on a scale of seconds. The game stoppage event is the referee marking a foul by whistle and arm signals or a ball being marked as out of bounds when it crosses the edge of the field. For goalie events that do not involve stop of game, those preceding events are usually saves of shot attempts, so the event of starting play is preceded by a goalie gaining control over the ball. Each analysis removes the observations with the upper and lower 5% of the event times in order to avoid outlier influence. We perform robustness checks that retain all observations and obtain the same results.

We use linear regression of the duration from the triggering event (play stoppage, ball out of bounds or ball captured by goalie) to play restart. The duration cannot be negative, so a censoring or event history model would have been possible. However, the distribution of the dependent variable shows that the boundary at zero does not cause observations to stack up at the lower bound, and hence a censoring model does not seem warranted. Nor are there cases

of right-censoring (i.e., no ending event). Referees stop for half-time and the end of game while the ball is in play, not while waiting for a team to restart the play, so a major rationale for using event history models is missing for these data. We thus chose linear regression as an estimator with good overall properties. We performed robustness checks using event history models, and found that our findings held.

We also analyze total time-wasting during a game by modeling the effective playing time as a function of time-wasting incentives as a linear regression.

### Variables

The independent variables for the team restarting the game are chosen to match the value-based and semi value-based time-wasting strategies. Our value-based model is parametrized using the betting odds as basis for the pre-match predictive distribution of team strengths. Prediction markets have been examined for bias and precision before, and found to have excellent properties in general [8, 9] and for soccer games [10, 11]. We obtain the odds from http://www.football-data.co.uk/, a site with odds for many leagues. For each match, we use the odds for home/tie/away prior to the match start to generate the corresponding Poisson arrival rates for goals $\lambda_1$, $\lambda_2$.

We define the following variables representing the value of a time-wasting strategy. Value of a minute is the expected value of wasting a minute to the restarting team, and equals the expected points (3 points for win, 1 point for tie, 0 point for loss) for the team in focus one minute later with the same score minus its current expected points. As the value of wasting an additional minute increases, teams should waste more time. Value of a minute is negative if the score relative to the expected playing time implies that the team would be better off gaining rather than wasting time (i.e., a superior team is behind in the score). To check whether the value of a minute has different effects for delaying or speeding up the game, it is split into a negative and a positive part. The absolute value is taken in the negative range, so a higher value means further away from the origin in either direction. The split variable will capture whether teams have different ease of speeding up and slowing down the game. Different ease of speeding up and slowing down the game could occur because there is no penalty for rushing a restart except that the team-mates will have less time to position themselves optimally. Slowing down the game can attract referee attention and potentially be penalized.

To examine the semi value-based model with anchoring and adjustment, we define two variables. Initial advantage is equal to the expected points of the restarting team less the expected number of points of the opposing team, and is calculated at the start of the match using the betting odds. The longer the duration of a stochastic process, the smaller the ratio of randomness to expected value, which means that longer effective playing time makes a favorable outcome for the stronger team more likely. Hence a team with larger value of initial advantage will have less incentive to delay the game. Performance reflects the change in expected points from the start of the match until the current time in the match, taking time and score into account. A positive value of this variable means that the team is doing better than expected, so to finish the game while the fortunate position remains it has an incentive to waste time.

The difference between the variables value of a minute and performance is important. Value of a minute represents a nearly optimal time-wasting strategy because it measures, at each time point, how much the team stands to gain from wasting a minute. The only omission from the model is the yellow card risk, which is difficult to quantify because it usually does not affect the current game, but may cause a player to lose eligibility for a later game. Performance is not optimal, though it has a similar analytical foundation. It corresponds to a backward-looking aggregate change in expected points since the game started. Thus, initial advantage is

an anchor of the likely points the team will take; performance is an adjustment for the current state of the game. These variables correspond to a cognitive operation of anchoring judgments at a reference point and making adjustments thereafter [7].

We also estimate a heuristic model similar to the graphs in Fig 1. It has the difference in goals between the restarting team and the opposing team, broken into the intervals -3 (3 or more goals behind), -2 (two goals behind), -1 (one goal behind), 0 (tied), 1 (one goal ahead), 2 (two goals ahead), and 3 (3 or more goals ahead). It has an indicator for whether the home team is the one restarting, and indicators for the nominal time played so far, divided into 6 time intervals of 15 minutes of play time. The estimates of this model are available from the authors and show that (as expected) this model is a worse fit to the data than the value-based and semi value-based models, for all events.

For the model of effective playing time, we divide a match half into three time buckets ($[0-15]$, $[15-30]$, $[30-end]$). For each time bucket, we calculate positive incentives as mean (max(0, Value of minute)) for either the home or away team, and negative incentives as mean (max(0, -Value of minute)) for either the home or away team. In some models, we also enter fixed effects for each team.

## Results

We display the model estimates in Table 1. In the top panel, the signs of all coefficient estimates match the prediction of the value-based model of time-wasting. Higher value of a minute leads to greater time-wasting when the value of a minute is greater than zero. When the value of a minute is below zero, the further below zero it is, the less time is wasted. This finding is remarkably consistent across the many types of stops modeled, especially when considering the great variation in players involved in each case. The two first outcomes are controlled by

**Table 1. Models of time-wasting.**

| Outcome | Goalie restart play | Goalie kick in play | Throw-in | Free kick | Corner kick | Substitution |
|---|---|---|---|---|---|---|
| *Value-based model* | | | | | | |
| (Intercept) | 25.436*** (0.040) | 6.590*** (0.032) | 12.009*** (0.024) | 26.936*** (0.077) | 25.033*** (0.052) | 50.52** (0.241) |
| Value of minute, positive | 129.637*** (6.079) | 165.078*** (5.021) | 187.904*** (4.363) | 315.646*** (12.904) | 189.899*** (10.602) | 195.340*** (26.936) |
| Value of minute, negative | -407.510*** (7.539) | -175.778*** (5.300) | -180.017*** (3.765) | -208.627*** (12.532) | -226.028*** (7.147) | -291.373*** (30.989) |
| AIC | 211,970 | 181,936 | 543,524 | 406,259 | 131,795 | 75,393 |
| *Semi value-based model* | | | | | | |
| (Intercept) | 24.397*** (0.029) | 6.363*** (0.023) | 11.879*** (0.017) | 26.972*** (0.055) | 24.808*** (0.038) | 49.851*** (0.169) |
| Initial advantage | -0.761*** (0.024) | -0.724*** (0.019) | -0.475*** (0.015) | -0.868*** (0.047) | -0.596*** (0.031) | 0.325** (0.141) |
| Performance | 2.506*** (0.038) | 1.500*** (0.030) | 1.757*** (0.022) | 2.473*** (0.071) | 2.130*** (0.047) | 2.787*** (0.168) |
| AIC | 211,206 | 181,065 | 542,133 | 405,934 | 131,253 | 75,323 |
| Observations | 34,238 | 32,119 | 89,332 | 51,504 | 21,247 | 8,981 |

Note:

***p<0.001;

**p<0.01;

*p<0.05; two-sided tests.

Standard errors in parentheses below coefficient estimates. AIC is the Akaike Information Criterion for model fit (lower means better model).

the goalie, throw-ins are often done by players in side-back or wing positions, and free kicks and corner kicks are typically done by the strongest and most accurate kickers. For some outcomes the coefficients for below-zero value are larger than those for above-zero value, suggesting that it could be easier to speed up play than to slow down play. The finding points to a role of the referees in countering the time-wasting strategy, as slowing down the game can be penalized but speeding up is not rewarded.

Interpretation of the coefficients is easy. If wasting a minute gives (say) a 0.1 increase in the expected points from the game, a goalie will waste 12.96 seconds when starting play with a goal-kick. This is a very large effect, but such benefits are found in the data. However, the value of a minute has a standard deviation of 0.007, so it is more normal to expect goal kick delays of just below two seconds, corresponding to a two standard deviations difference. This is already noticeable for spectators (especially fans of an opposing team that is behind), and the model accurately captures spectators' impression that goalies waste time when it benefits their team. Other events such as free-kicks have much greater coefficients, indicating that they are more amenable to time-wasting. Part of this may be explained by the fact that a free kick is often due to foul play that can result in temporary pain. It is hard to assess how the fouled player is affected, so he has the opportunity to exaggerate the impact. A two standard deviation difference in the value of a minute predicts more than four seconds added when taking a free-kick. The value-based models have good fit to the data, but comparison of the AIC statistics with the semi value-based models in the bottom panel shows that the semi value-based models fit the data best.

In Table 1, bottom panel we see negative coefficients for the initial advantage, showing that stronger teams do not tend to delay the game, which is expected because a stronger team is in a better position to score, and needs to score in order to win the game. It is also more likely to be the one scoring when ahead in order to secure the win. Also, the coefficient for performance has a positive sign, indicating that a team that does better than expected will waste time. This coefficient is significant across all the events we analyze, in support of our prediction of strategic time-wasting.

These differences are large enough to be visible in simple graphical displays. Fig 2 shows box plots for the duration of four restart events grouped into low (bottom 10%), medium (center 80%), and high (top 10%) game-time performance. Compared with the analysis, this summation of the data gives an incomplete view of the strategic time-wasting because the initial advantage also affects the delay and varies among the observations in each of the groupings. Still, the graphs show clear effects for all outcomes, with the median increasing exactly as the theory would expect in all three graphs, the first quartile increasing in all four graphs, and the third quartile increasing in all graphs except free kicks. The proportional increase is remarkably large for throw-ins, whereas free kicks see a large absolute increase but much smaller proportional increase due to the many observations with very long duration. For corner kicks, both the proportional and absolute increases are large. Graphs dividing the data in percentiles of value of a minute showed similar effects.

Next, we estimated the total effect of the time-wasting incentives on the effective playing time. The findings are in Table 2. We see that the R square statistic is 0.091 in Model 3 before the fixed effects for teams are entered, which means that the full set of time-wasting incentives explains 9.1% of the time lost. This explanatory power is good considering the significant variation in random sources of loss in effective playing time loss, such as actual (not faked) player injuries. The significant increase in R square in Model 4 indicates that teams also differ greatly in time-wasting, and by inspecting the coefficients of the team indicator variables it is easy to find pairs of teams whose expected effective playing time differ by at least five minutes per game. As one might expect, the team fixed effects are related to the team strength. Fig 3 shows

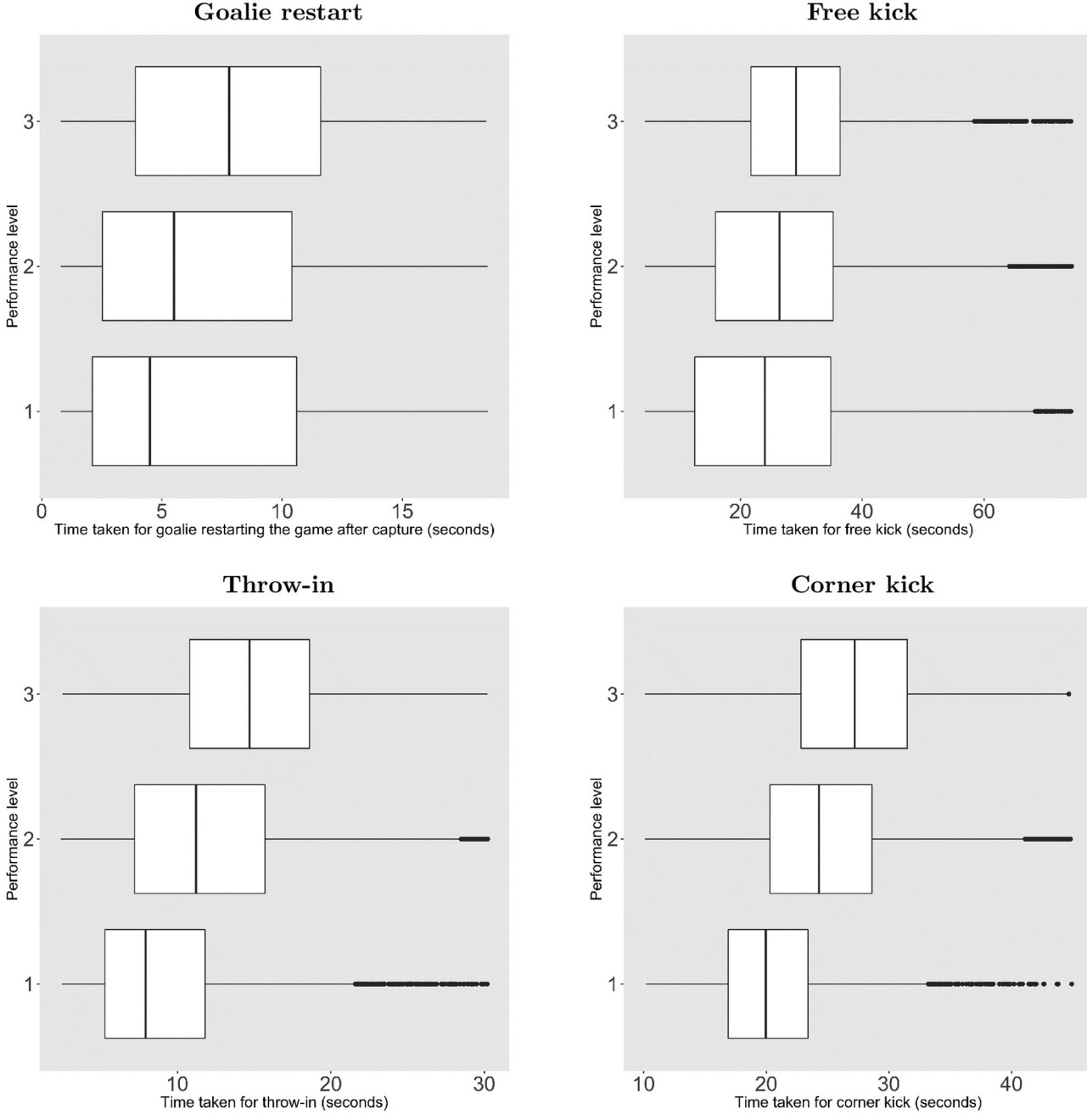

**Fig 2. Delay of game as function of performance. Performance level**. 1 = Bottom 10% (low performance). 2 = Middle 80%. 3 = Top 10% (high performance).

scatterplot of average effective playing time per home game by average points by home game (green triangles) and away game (red circles) with fitted lines. Teams are awarded zero points for a loss, one for a tie, and three for a win. The fitted lines shows greater effective playing time for teams that earn more points per game, as one would expect if time-wasting is a way for weak teams to seek wins against stronger teams. The dispersion of expected effective playing time is signficant, both on an absolute scale and relative to the fitted line. The large effects of time-wasting incentives and the large interteam variation suggests that rule changes that

**Table 2. Systematic components of effective playing time.**

| Variable | Model 0 | Model 1 | Model 2 | Model 3 | Model 4 |
|---|---|---|---|---|---|
| Incentives home | | yes | | yes | yes |
| Incentives away | | | yes | yes | yes |
| Home fixed effects | | | | | yes |
| Away fixed effects | | | | | yes |
| Observations | 2231 | 2231 | 2231 | 2231 | 2231 |
| Residual std. errors | 297.143 | 291.159 | 287.172 | 284.780 | 222.285 |
| $R^2$ | 0 | 0.045 | 0.071 | 0.091 | 0.461 |
| F-statistic | - | 8.717*** | 14.129*** | 9.242*** | 21.892*** |
| Df | - | 12 | 12 | 24 | 84 |

Note:

***$p<0.001$;

**$p<0.01$;

*$p<0.05$; two-sided tests.

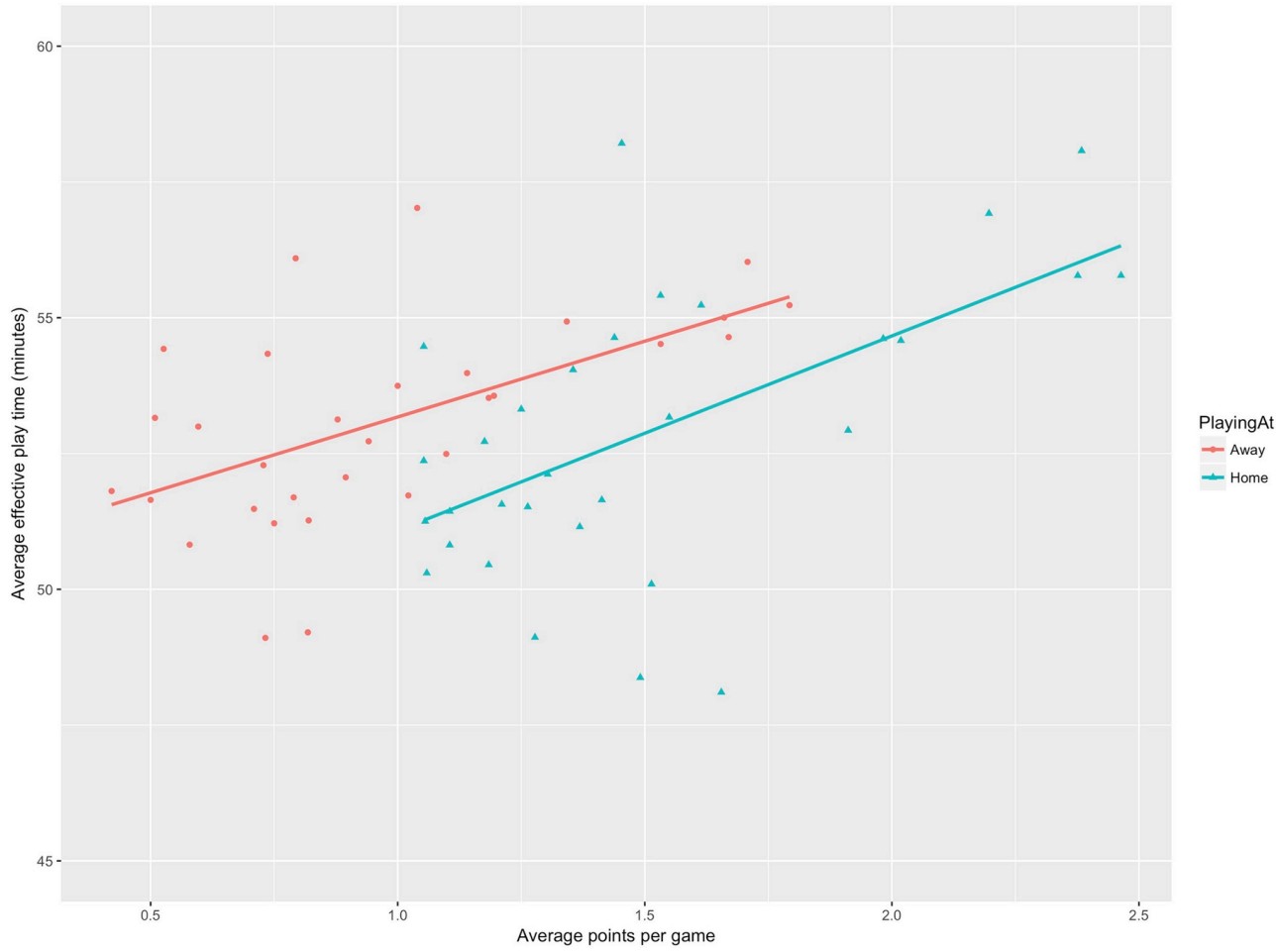

**Fig 3. Average effective playing time by average points at home and away.**

control time-wasting would be valuable. Whether stopping the clock when the ball is out of play is the best solution cannot be determined from these findings, but it is clear that the norm of fair play and the adherence to rules against strategic rule-breaking are under threat.

Our model estimates can give some indication of how stopping the clock when the ball is out of play might affect teams and audiences. We estimated the time taken to restart the game when teams had a positive and negative value of minute and compared it with the estimated time they would have taken if the value of a minute was zero. The calculations show that for the audience the rule change would have speeded the game for every outcome except corner kicks, in which the time saving by teams rushing the play was 48 percent higher as compared to the teams slowing down the play and this time saving would be lost when there is no incentive to rush or slow down the play, and throw-ins, which only had 1 percent more time spent for teams wasting time. Goalie restart was 24 percent slower, goalie kick was 80 percent slower, freekick was 36 percent slower and substitutions seven times slower for teams seeking to waste time. For the audience, we would thus expect the effect of a rule change to be faster restart in some situations and slower in others, but overall more effective playing time. Indeed, if scoring rates were unchanged, the total time saving would equal one more goal per 11 games.

For teams, a fast restart is not necessarily good either. Restart of play is best done when players have found good positions on the field, which can take some time. It is possible that the every restart of the game by teams with a negative value of a minute is done at exactly the right time tactically, but it is also possible that some of these restarts are done too soon and hence place the team at a disadvantage. From the viewpoint of the soccer governing bodies, fair play and entertainment of fans are central goals. This means that time wasting is problematic, and even time saved by teams at a disadvantage is problematic because it violates the norm of fair play.

## Conclusion

The findings showed remarkable consistency and clarity. Time-wasting is illegal, subject to referee penalty, and countered by opposing team attempts to hurry up the restart of play. Nevertheless, the models showed that strategic time-wasting is widespread. The findings were consistent across many time-wasting opportunities. The empirical results and simulations offer clear advice to the soccer governing bodies. The goal of having a fair game with limited strategic rule-breaking requires a different set of rules, for example by stopping the game clock when the ball is out of play. What set of rules will be best needs to be determined through a combination of further modeling and actual game-play. Changing the time-keeping rules of soccer is a major change of rules in a sport that has very broad fan base and great respect for traditions, and should be tested before taking final action.

The main theoretical result is that a value-based model describes the behaviors less well than a model that takes into account the benefits of time delay, but does so through anchoring and adjustment rather than as an optimal response to the benefit of each minute of delay. This is true even though the team benefits from the rule-breaking, but the enforcement of the rules are directed towards punishing the individual player. The findings show the limitations of creating rules, because individuals can employ strategies that go against the rules and benefit from rules not being perfectly enforced, as in the time-wasting strategies studied here. Given sufficient encouragement and rewards from their team and its fans, they are willing to do so, and they execute the rule-breaking with significant skill.

Indeed, a second major result is the high predictive power of the value-based model and the semi value-based derived from similar variables. This does not mean that soccer players estimate the difference of two Poisson distributions in their heads, as these models assume.

Rather, they know the scores and times that most reward time-wasting and can make fine-tuned adjustments in response to this knowledge. After all, their experience playing the game is substantial, and the adjustments they make based on this experience fit the predictions of these models very well, even though the models involve calculations that go beyond what one would expect a soccer player (or professor) to be able to do on the fly.

Our investigation suggests future lines of research. Increasingly, finer grained data on decision making in many contexts are becoming available, especially for decision making in sports. They can reveal the tension between rules and rule-breaking incentives, and the tension between organizational goals and individual rewards. The data can reveal how an individual performs for the organization, including (as in this study) rule-breaking to benefit the organization. Further analysis can show what situations strengthen or weaken the optimality of decision making. There are good opportunities to further explore such data for strengthening the evidence on the decision-making strategies and the resulting quality of individual decision making for organizations and to analyze how rules affect behavior and hence inform entities responsible for making rules that control individuals and organizations.

## Acknowledgments

We are grateful to Amisco-Prozone for giving us access to their data. This paper has benefited greatly from Petter Rudi sharing his experience as a player in Premier League and Italy's Serie A. Kjetil Siem at FIFA and Lukas Brud at IFAB have provided helpful discussions, and seminar participants at INSEAD have given helpful comments. The authors are listed alphabetically.

## Author Contributions

**Conceptualization:** Henrich R. Greve, Nils Rudi, Anup Walvekar.

**Data curation:** Henrich R. Greve, Nils Rudi, Anup Walvekar.

**Investigation:** Henrich R. Greve, Nils Rudi, Anup Walvekar.

**Methodology:** Henrich R. Greve, Nils Rudi, Anup Walvekar.

**Writing – original draft:** Henrich R. Greve, Nils Rudi, Anup Walvekar.

**Writing – review & editing:** Henrich R. Greve, Nils Rudi, Anup Walvekar.

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
