## [Decision Letter · Decision Letter 0]

18 Aug 2019

PONE-D-19-19053

Rational rule breaking: Time wasting to win soccer games

PLOS ONE

Dear Dr. Greve,

Thank you for submitting your manuscript to PLOS ONE. After careful consideration, we feel that it has merit but does not fully meet PLOS ONE’s publication criteria as it currently stands. Therefore, we invite you to submit a revised version of the manuscript that addresses the points raised during the review process.

Please find below the reviewers' comments, as well as mine.

We would appreciate receiving your revised manuscript by Oct 02 2019 11:59PM. To enhance the reproducibility of your results, we recommend that if applicable you deposit your laboratory protocols in protocols.io, where a protocol can be assigned its own identifier (DOI) such that it can be cited independently in the future. For instructions see: http://journals.plos.org/plosone/s/submission-guidelines#loc-laboratory-protocols

We look forward to receiving your revised manuscript.

Kind regards,

Valerio Capraro

Academic Editor

PLOS ONE

Journal Requirements:

Additional Editor Comments:

I have now collected two reviews from two experts in the field. Both reviewers like the paper but their overall judgments are somewhat split: one recommends acceptance and one recommends major revision. Therefore, I would like to invite you to revise your paper following the comments suggested by one of the reviewers.

Looking forward for the revision.

Reviewers' comments:

Reviewer's Responses to Questions

**Comments to the Author**

1. Is the manuscript technically sound, and do the data support the conclusions?

Reviewer #1: Partly

Reviewer #2: Yes

2. Has the statistical analysis been performed appropriately and rigorously? 

Reviewer #1: Yes

Reviewer #2: Yes

3. Have the authors made all data underlying the findings in their manuscript fully available?

Reviewer #1: No

Reviewer #2: Yes

4. Is the manuscript presented in an intelligible fashion and written in standard English?

Reviewer #1: Yes

Reviewer #2: Yes

5. Review Comments to the Author

Reviewer #1: Soccer is going through its own “Moneyball” moment, as teams make use of extraordinarily detailed data on player movements and game play to optimize their strategies, see e.g., “The Numbers Game” by Chris Anderson and David Sally. To the best of my knowledge, this is the first paper in the management-science literature to use that same data.

The authors consider stoppages (due to throw-in, free kick, goalie restart, etc) and show convincingly that teams that are leading the game tend to take more time to restart while teams that are losing take less time to restart, compared to when the score is tied. Moreover, the magnitude of these effects is substantial. For instance, when throwing in the ball, teams that are leading take about 15% more time than teams that are tied, who in turn take about 15% more time than those that are losing. This is interesting.

Unfortunately, I have three major concerns with the paper that make it unsuitable for publication in its current form.

FIRST, the authors’ key conclusions do not follow from their findings. “We predict that a change to rules with stopped game clock when the play is stopped would make game play more time efficient” (abstract). This would be true if “strategic time-wasting” was the only effect at play but, as noted above, teams that are losing ALSO respond to their incentives, speeding up play in their urgency to score before time runs out. So, any time savings that one would get by speeding up the leading team will be counteracted by slowing down the trailing team. Indeed, to the extent that the trailing team benefits more from resting its players—to be capable of a few last bursts of athleticism to score—we might expect the losing team to delay even MORE than what the analysis here suggests. Overall, then, it is unclear whether stopping the clock would reduce or increase the amount of time wasted overall. Certainly, it is NOT true that “The empirical results and simulations offer clear advice to the soccer governing bodies” (conclusion).

To resolve this first concern, the authors should provide a balanced analysis that emphasizes how the current rules incentivize both time-wasting and time-saving. They could then discuss how switching to a stopped clock would have mixed effects (speeding return to play in some cases, slowing it in others) that need to be carefully weighed.

SECOND, the authors’ discussion of causation of patterns found in the data is questionable and likely to confuse readers who are unfamiliar with causal reasoning. For example: “Teams that are ahead get fewer opportunities to throw the ball in, consistent with the opposing team seeking to deny them this time-wasting opportunity.” The authors here are suggesting a fascinating possibility, that trailing teams might be intentionally keeping the ball in bounds, to stop leading teams from wasting time. But the paper provides no causal analysis to support this possibility. This could leave readers with the wrong impression that the conjectured causal explanation has been shown to be true. (Experts understand that the phrase “consistent with” is code for “we have no idea if this is true”, but this paper could reach a broader audience of soccer enthusiasts who are not as familiar with the terminology.)

This second concern can be resolved by dropping all causal speculation or, if the authors want to discuss potential explanations, to be sure to include at least two alternative explanations for each pattern discussed.

THIRD, the “rational model” is not really a “rational model”. The **value** of delay is introduced in line 185 and corresponds to the extra likelihood of winning or tying when there is less time remaining (which, of course, depends on the current state of the game, team characteristics, etc). However, there is no **cost** of delay! (In the case of slowing down by a leading team, the benefit is that having less time remaining improves odds of winning, while the cost is that being penalized decreases odds of winning. In the case of speeding up by a trailing team, the benefit is that having more time improves odds of tying, while the cost is that rushing may lead to a poorer re-start to play.) Moreover, the extent of these costs are undoubtedly non-linear in the amount of delay, e.g., referees may be more likely to give a yellow card as more time passes without a restart of play. (The authors appear to obliquely reference cost in lines 195-196 “The split variable will capture whether teams have different **ease** of speeding up and slowing down the game.” However, I find this very confusing since the “split variable” is referring to different “parts” of the value of a minute, meant to capture “the different effects for delaying or speeding up the game.” What does that mean??)

The term “rational” refers to a choice being made optimally in pursuit of some objective. To study a “rational” actor, one must therefore fully specify the objective that is being maximized. For instance, to capture the objective of the leading team and hence have a truly rational model of delay here, one must not only estimate the “value of a minute” (as done, capturing the benefit of delay) but also the “cost of a yellow card” and “likelihood of a yellow card” (capturing the cost of delay).

This critique may seem like semantics, but there is important substance to it as well. With a rational model, one can explore and make predictions about what would have happened if the game had been changed. For instance, what if soccer’s governing bodies decided to encourage referees to increase the frequency with which they give yellow cards for delay. Armed with a rational model, one could conduct the relevant counterfactual analysis and produce quantitative estimates about the likely impact of such an intervention. By contrast, based on this paper’s analysis, you could say that there will be less delay—but have no way of quantifying how much less.

This concern could be addressed by motivating the writeup and framing the analysis more accurately, namely, as a calibration exercise, e.g. asking: “How much does delay increase or decrease winning chances [i.e. what is impact of delay]?”, “How much do teams respond per extra-% of winning by delaying the return to play [i.e., how sensitive are teams to the impact of delay]?”, and “Is each team’s sensitivity consistent across contexts, i.e., it is similar for throw-ins as for free kicks?”

Switching now from concerns to suggestions, here is an idea for a different sort of policy intervention: Introduce a new penalty “PURPLE CARD” that is issued specifically for delay in the latter part of the game (when strategic delay is most likely) and, when issued, gives the coach of the other team the option to extend game-play for one extra minute per purple card issued. A player on the leading team who drags his feet a few extra seconds making a throw-in now risks giving the other team one whole minute of additional time. Such a policy would keep the excitement of the trailing team’s urgency to get the ball back into play as soon as possible, while also reducing the leading team’s incentive to delay the game. (Another advantage of such a rule change is that it could be easily scaled, by adjusting the stringency with which referees enforce the rule. By contrast, eliminating clock stoppages is a dramatic and irreversible change—or, if it is reversed, highly embarrassing for the governing body!!)

Smaller comments

1. Line 69-70: Please check the sentence after “Corners show the same pattern”. Does “less” need to be switched to “more”? The pattern described appears to be the opposite of the pattern for free kicks.

Reviewer #2: The paper is done responsibly, though I have a couple comments.

1) I'm not sure rational vs. non-rational is the right terminology. I know the spirit of rational expectations is that rational means forward looking, but this applies in specific contexts like asset pricing. Rational behavior could mean using past or current information in this context.

2) Your model with Poisson arrival suggests non-linear behavior. You might consider using more sophisticated estimation techniques in future work.

6. PLOS authors have the option to publish the peer review history of their article (what does this mean?). If published, this will include your full peer review and any attached files.

Reviewer #1: Yes: David McAdams, Professor of Economics, Duke University

Reviewer #2: No

---

## [Author Response · Author response to Decision Letter 0]

1 Oct 2019

Reviewer 1: Thank you for your comments. We have revised our manuscript accordingly, and describe the changes in detail in our attached response to reviewers. 

Reviewer 2: Thank you for your comments. We have revised our manuscript accordingly, and describe the changes in detail in our attached response to reviewers.

---

## [Decision Letter · Decision Letter 1]

8 Oct 2019

Rational rule breaking: Time wasting to win soccer games

PONE-D-19-19053R1

Dear Dr. Greve,

We are pleased to inform you that your manuscript has been judged scientifically suitable for publication and will be formally accepted for publication once it complies with all outstanding technical requirements.

With kind regards,

Valerio Capraro

Academic Editor

PLOS ONE

Additional Editor Comments (optional):

Reviewers' comments:

Reviewer's Responses to Questions

**Comments to the Author**

1. If the authors have adequately addressed your comments raised in a previous round of review and you feel that this manuscript is now acceptable for publication, you may indicate that here to bypass the “Comments to the Author” section, enter your conflict of interest statement in the “Confidential to Editor” section, and submit your "Accept" recommendation.

Reviewer #1: All comments have been addressed

2. Is the manuscript technically sound, and do the data support the conclusions?

Reviewer #1: Yes

3. Has the statistical analysis been performed appropriately and rigorously? 

Reviewer #1: Yes

4. Have the authors made all data underlying the findings in their manuscript fully available?

Reviewer #1: No

5. Is the manuscript presented in an intelligible fashion and written in standard English?

Reviewer #1: Yes

6. Review Comments to the Author

Reviewer #1: I am satisfied with the revision. Small detail: typo on pg 10, line 32, "every restart" should be "early restart"

7. PLOS authors have the option to publish the peer review history of their article (what does this mean?). If published, this will include your full peer review and any attached files.

Reviewer #1: Yes: David McAdams

---

## [Editor Report · Acceptance letter]

22 Nov 2019

PONE-D-19-19053R1 

Rational rule breaking: Time wasting to win soccer games 

Dear Dr. Greve:

I am pleased to inform you that your manuscript has been deemed suitable for publication in PLOS ONE. Congratulations! Your manuscript is now with our production department. 

With kind regards,

on behalf of

Dr. Valerio Capraro 

Academic Editor

PLOS ONE